# Positive or Negative Viewpoint Determines the Overall Scenic Beauty of a Scene: A Landscape Perception Evaluation Based on a Panoramic View

**Yue Chen** †, **Qikang Zhong** †  and **Bo Li** *

School of Architecture and Art, Central South University, Changsha 410083, China; chenyue0905@126.com (Y.C.); 201311063@csu.edu.cn (Q.Z.)
* Correspondence: libo0910@csu.edu.cn
† These authors contributed equally to this work.

**Abstract:** In the contemporary world, the swift advancement of urbanization, the pressing need for environmental conservation, and humanity's unyielding quest for a better quality of life have jointly underscored the escalating importance of research on landscape aesthetics and perceptual experiences. Researchers have often evaluated the overall scene's beauty based on photos taken from a single viewpoint. However, it has been observed that different viewpoints of the same scene can lead to varying degrees of beauty perception. Some positive viewpoints highlight landscape features that contribute to beauty preferences, while negative viewpoints emphasize aspects that may evoke discomfort and decrease perceived beauty. Therefore, a crucial question arises: which viewpoint, positive or negative, holds more influence over the overall beauty of the scene? This paper aimed to address this question by utilizing panoramic map technology to establish a landscape perception evaluation model. The model was based on empirical evidence from various spatial scenes along the Yaozijian Ancient Road in Anhua County, encompassing towns and villages. The study analyzed the functional relationship between landscape factors, positive and negative viewpoints, and the degree of scenic beauty. It was found that (1) it is difficult to reflect the overall scenic beauty of a scene (OSBS) of a single viewpoint photo, and both positive and negative viewpoints of scenic beauty have significant effects on the OSBS. In the empirical case study, it was found that the overall effect of a positive viewpoint of scenic beauty (PVSB) on OSBS was greater; (2) PVSB had a major effect on OSBS with a high visual hierarchy and cloud ratio and a low type of vegetation and proportion of man-made objects; (3) a negative viewpoint of scenic beauty (NVSB) had a major effect on OSBS with a low visual hierarchy of the landscape. The results of the study reveal the relationship between landscape factors of different viewpoints and the OSBS, which can be applied to landscape beauty evaluation and landscape planning and design processes.

**Keywords:** overall scenic beauty; landscape perception evaluation; positive landscape factor; negative landscape factor; panorama

## 1. Introduction

In today's globalized world, landscape perception evaluation as an important research area has attracted widespread attention among different fields and stakeholders. In the contemporary world, the rapid expansion of urbanization has ushered in profound transformations in urban landscapes, triggering substantial shifts in people's lifestyles and living environments. Against this backdrop, the evaluation of urban landscape perception goes beyond mere considerations of the quality of life and well-being of urban residents; it is intricately linked to the sustainable development and societal harmony of cities. A comprehensive investigation into urban landscape perception evaluation holds promise for uncovering innovative approaches to forging livable, enchanting urban environments and elevating the satisfaction and sense of belonging amongst city dwellers. Furthermore,

environmental conservation has emerged as a pressing global concern, and landscape perception evaluation plays a paramount role in this narrative. Gaining insights into people's perceptions and evaluations of natural landscapes is pivotal in underscoring the positive impact of natural environments on human health and well-being. Simultaneously, the preservation and restoration of natural landscapes necessitate due consideration of public perceptions and evaluations, necessitating amplified public engagement to drive the pursuit of sustainable environmental conservation. Moreover, landscape perception evaluation transcends the realm of individual environmental perceptions, delving into their influence on the broader aesthetics and emotional experiences. Embracing a comprehensive perspective, landscape enmeshes subjective evaluations and perceptual encounters concerning natural landscapes, urban settings, and architectural constructs. Unearthing people's perceptions and evaluations of distinct landscape typologies can unveil the psychological and emotional ramifications of diverse environments for humanity, bestowing invaluable theoretical guidance to craft more captivating and comforting landscapes. In this vein, the assessment and enhancement of urban landscape aesthetics and perceptual experiences emerge as pivotal imperatives in the domains of urban planning and landscape design.

In recent years, important progress has been made in the study of landscape beauty and perceptual experience in the fields of urban planning [1–3], environmental design [4–6], tourism planning [7,8], and psychology [9–12]. In the current research landscape, the primary focus lies in investigating the impact of emotional experiences, environmental features, and socio-cultural aspects on the evaluation of landscape aesthetics. Concurrently, these studies grapple with intricate subjective factors, giving rise to noteworthy variations in perceptions and evaluations among individuals of diverse cultural backgrounds. Indeed, the subjective experiences of beauty and comfort in natural and architectural environments play a pivotal role in shaping how individuals assess and interact with their surroundings. In this pursuit, researchers employ a diverse array of methodologies, encompassing survey questionnaires, rating techniques, on-site observations, and cutting-edge advancements in neuroscience. This multifaceted approach enables a comprehensive understanding of landscape perception evaluation, delving into the intricate interplay of cognitive and emotional responses in shaping individuals' aesthetic judgments. Furthermore, within the realm of landscape perception evaluation, the linkage between different landscape elements (ranging from natural landscape features, urbanization aspects, environmental quality, and conservation considerations to visual hierarchy and organizational components and even cultural diversity factors) remains a critical and stimulating research frontier. Through an in-depth exploration of the influence wielded by these diverse landscape elements on aesthetic perceptions and emotional experiences, researchers have unlocked valuable insights that resonate deeply with landscape designers and urban planners, offering indispensable guidance and inspiration. In conclusion, the ongoing quest to comprehend landscape aesthetics evaluation delves into multifaceted dimensions, from understanding human emotions and societal dynamics to employing diverse research methodologies. The findings from these studies are reshaping our understanding of landscape perception and propelling the creation of harmonious, appealing urban environments that foster a sense of connectedness and appreciation among inhabitants. However, in past studies, landscape perception evaluation usually relied on photographs or 2D images, ignoring the influence of different viewpoints. Different perspectives of the same scene can trigger different degrees of beauty, in fact, and this choice of perspective is often closely related to beauty preferences and emotional responses. Some perspectives reflect positive landscape factors that determine beauty preferences, such as open landscape views and the presence of natural elements, while other perspectives may highlight negative landscape factors that lead to beauty discomforts, such as visual clutter and environmental oppressiveness. Therefore, exploring the relative contribution of positive and negative perspectives (referred to as capturing the most beautiful aspect and the least appealing aspect within a panoramic shot) to the overall scenic beauty of a scene (OSBS) becomes an important and pending issue for further research.

To more accurately assess and predict the OSBS, emerging panoramic mapping technologies provide a more realistic and immersive perceptual experience. Traditional landscape studies typically rely on photographs or 2D images to depict scenes [5,6,12,13]. However, this approach falls short in recreating the genuine feelings and experiences of people within the actual environment. By capturing image information from multiple viewpoints, the panorama technique provides a broader and more lifelike view of the surroundings, creating a more authentic experience for observers. As a result, there has been considerable research interest in exploring the potential of this technique for landscape perception evaluation. However, current studies on the specific application of the panorama technique in landscape perception evaluation are mostly focused on a certain type of landscape empirical evidence, such as natural landscapes and architectural landscapes [5,6,12]. As a result, significant knowledge gaps persist regarding the relationship between different viewpoints and the varying influence of landscape factors on beauty perception. Some recent scholars have recently started to use panorama techniques to investigate the effects of different scenes on landscape perception evaluation. For example, Chen et al. (2022) found that the contribution of natural landscape elements and public space structures to rural aesthetics varied across scenes [2]. However, these studies still do not reveal the mechanisms of the influence of positive and negative perspectives on OSBS. Especially in the context of incorporating panoramic mapping techniques, there is a need to delve deeper into the functional relationship between different perspectives and scenic beauty degrees. Understanding the mechanisms through which landscape factors influence scenic beauty under various viewpoints is crucial.

Therefore, the objective of this study was to develop a landscape perception evaluation model based on panoramic map technology using the Yaozijian Ancient Road in Anhua County as an empirical object and to analyze the relationship between positive and negative viewpoints and the scenic beauty degree. Through systematic empirical research, this study explored the relative importance of different viewpoints in the OSBS and the degree of contribution of landscape factors to the beauty under different viewpoints. This will help deepen the understanding of landscape perception evaluation and provide a scientific basis to guide landscape planning and design.

## 2. Review

### 2.1. Advantages and Limitations of Traditional Photos and 2D Images in Visual Evaluation of Landscapes

The visual evaluation of landscapes holds significant importance for researchers in landscape planning and design. Traditional photographs and 2D images are widely used as evaluation tools in this field. They have been extensively applied in various studies, including visual evaluation [2,12,14,15], preference surveys [1,5,6,8], affective assessment [16–18], and simulated experiences [1,3,16,19]. For example, Tang et al. (2015) employed photographs and images, prompting participants to assess the quality of three different types of rural forest landscapes based on their personal feelings and subjective evaluations. These assessments enabled researchers to compare differences between landscapes [12]. Luckmann et al. (2013) presented participants with various landscape photographs or images, requesting them to rank, score, or select photos according to their individual preferences. This approach allowed researchers to collect data on individual landscape preferences [6]. Petrova et al. (2015) asked participants to view photos or images and assess them on emotional dimensions, yielding similarities and differences in visual and emotional assessments of landscapes based on ethno-cultural and regional differences [18]. Chen et al. (2022b) provided participants with real or virtual landscape images, allowing them to mentally experience specific landscape environments, and subsequently assessed the effects of landscape on individual psychological and emotional states [16].

These related studies reveal the advantages of the controllability, reproducibility, and generalizability of photographs or 2D images in landscape perception assessment. Scholars such as Qin et al. (2023) emphasize that utilizing photographs and geospatial

data for landscape visual quality assessment offers controllability, enabling direct comparisons between scenes and delivering consistent quantitative and qualitative results [1]. Munoz-Pedreros (2004) noted that using photographs as an assessment tool can increase efficiency and reduce participant workload, thereby reducing the cost and complexity of surveys [13]. Svobodova et al. (2018) and others used photographs as stimulus material to compare and measure participants' visual evaluations with natural values. By using photographs, researchers were able to ensure that each participant saw the same landscape stimuli, eliminating possible variability in field surveys [20]. Chen et al. explored the use of photographs and images to visualize and measure changes in urban green spaces. Employing photographs and images allows researchers to conduct assessments in a safe and controlled environment, especially for landscape types that may be potentially risky or inaccessible in the field [3,21,22]. Akten and Celik et al. (2013) assessed the visual quality of landscapes in urban parks by using photographs. Studies have shown that photographs can provide reliable visual material to compare and analyze the visual quality of landscapes in different geographical and cultural contexts [8]. Photographs or two-dimensional images substantially help existing studies, however, some researchers have raised concerns about the validity of photo evaluation techniques. It has been pointed out that individual photographs may fully capture the diversity and dynamics of the scene, leading to potentially inaccurate evaluation results. Since photographs from different viewpoints have different degrees of beauty, using a single photograph may not accurately reflect the observer's beauty evaluation of the overall scene [1,20,22,23]. Moreover, traditional photographs or 2D image evaluation techniques may not accurately convey the realism and emotional experience of the landscape; photographs do not provide a 3D and immersive experience and may not fully capture the overall scenic beauty of the landscape [2,24–26]. In the realm of panoramic imagery, the direct observation of two-dimensional photographs differs significantly from the immersive experience afforded by alternative methods, such as utilizing the "computer screen + indoor large screen projection" setup or engaging with VR devices to perceive 720° VR panoramas. These distinctions give rise to variations in comfort, perceptual experiences, and their suitability for diverse contexts. Specifically, the adoption of the "computer screen + indoor large screen projection" approach to view and experience 720° VR panoramas may present heightened levels of comfort and convenience. Consequently, the choice of viewing method may exert distinct influences on the perception and experiential aspects of the showcased landscapes. Importantly, observing panoramas on a screen may resonate more closely with respondents' inherent perceptual habits of the real world, primarily due to their preference for engaging with images and videos on curved displays. As the discourse surrounding the most optimal method for immersive panoramic experiences continues to evolve, an in-depth exploration of the interplay between perceptual habits and viewing technologies sheds invaluable light on the frontiers of panoramic landscape exploration. In addition, because photographs are selected and composed by the photographer, photo evaluation may be influenced by the subjective choices of the photographer and editing, and there may be inconsistencies in individual preferences and assessment results.

In conclusion, traditional photographs or 2D images offer certain advantages, such as controllability, cost-effectiveness, and safety, making them valuable tools in visual evaluation studies. However, they also come with inherent limitations, including a lack of foreground bias, limited field experience, static presentation, and subjective selectivity. Therefore, when utilizing these tools for evaluation, it is essential to carefully consider their drawbacks and complement them with other methods to achieve more comprehensive and accurate results.

### 2.2. Application of Panoramic Techniques in Landscape Perception Evaluation

The panorama technique is progressively gaining traction in landscape perception evaluation studies due to its comprehensive image presentation capabilities. Numerous studies have highlighted the evident advantages of using panoramas for landscape eval-



uation as they tend to stimulate observers' attention and evoke more positive emotions, resulting in enhanced reliability in beauty assessment [2,27–29]. Exploring the effects of different environments on observers' emotional experiences using panorama techniques has revealed that panoramas create a greater sense of relaxation, pleasure, and satisfaction among observers. Consequently, panorama techniques have proven to be a reliable means for emotional research [16,30,31]. In addition, panoramas can provide more details and visual information, enabling observers to perceive and evaluate the beauty of a scene more comprehensively. Chen et al. (2023) conducted a study using VR panoramas to present landscape scenes and discovered that panorama technology better portrayed the overall scenic beauty, garnering more interest and appreciation from observers [32–36]. Similarly, Sun et al. (2019) utilized VR panorama technology to present urban landscape scenes and extracted street vegetation coverage by using Sentinel-2 images and the greenscape index by using Baidu Street View panorama to assess the comprehensive evaluation index of street greening. The results of the study showed that the panorama technology can comprehensively measure the street greening distribution information and guide the planning and management of the urban greening landscape [37]. Several studies have also explored the application of panorama techniques in the evaluation of different types of landscapes [1,28,29,38,39]. For example, Zhang et al. used panoramic techniques to present natural landscape scenes to study participants' beauty evaluations of different landscape features (e.g., water, vegetation). The results of the study showed that the panorama technique could provide more diverse scene information, enabling participants to more accurately evaluate the impact of different landscape features on beauty [28,29,38]. Additionally, Qin et al. delved into the panorama technique's application in natural landscape perception. Utilizing panoramic images to showcase different natural landscapes, the researchers assessed participants' beauty experiences through subjective evaluations and physiological indicators, such as heart rate and electrical skin response. The findings indicated that panorama technology could provide a realistic landscape perception experience and capture participants' emotional and physiological responses [1,28,39].

Although the existing literature suggests that panoramas can reflect the OSBS, the beauty relationship between different viewpoints and the overall scene has not been adequately studied. Most studies have focused on assessing the OSBS of panoramas, while there is limited understanding of the effects of different viewpoints on the beauty experience and evaluation of the observer. Consequently, further research is necessary to explore the role of different viewpoints within panorama techniques and the observers' beauty preferences concerning these various perspectives. Such a study would be instrumental in enhancing our understanding of how observers develop an overall sense of scenic beauty, and it could offer more precise guidance for design and planning.

In summary, the existing literature suggests that panoramic technology has a wide potential for application in landscape perception evaluation. It can provide a more realistic and immersive observation experience and can more accurately capture participants' emotional responses and beauty evaluations. In addition, panoramic technology can help to study the influence of different environmental factors and perspectives on the OSBS, providing new research tools and perspectives. However, further in-depth research on the beauty relationship between different perspectives and the overall scene is needed to better understand and evaluate the overall beauty of the landscape.

## 3. Materials and Methods

### 3.1. Study Area

To analyze the mechanism of scenic beauty influence of different perspectives and landscape factors in different scenes and scenes of landscape tourism routes in urban and rural areas, historical walking trails through rural and urban areas were selected for empirical research. The study area is located in Huanghuaxi Village, Zhulinxi Village, Dongshi Village, Xitan Village, Sixian Village, Jiangnan Town, and Dongping Town in Anhua County, Yiyang City, Hunan Province, a total of two towns and five villages, which

include the Yaozijian Ancient Road and the modern towns within the radiation of its tourism supporting services. Yaozijian Ancient Road is an important starting point of the Wanli tea road (Hunan section) inscription, which is about 30 km from the Yuanqi Bridge in Huanghuaxi Village, Jiangnan Town, Anhua County, to the Wufu Gong Pier on the bank of Zijang River in Jiangnan Town, with a summit elevation of about 512 m. The main nodes of the ancient road from south to north are Yuanqi Bridge—Dapingxiehuo Store—Yaozijian Stone Road (Ganlu Pavilion, Yizhong, ancient monuments, cliff carvings, Meng Gong Temple)—Juegong Bridge—Dongsi Old Street—Yongxi Bridge Yongxi Bridge)—Sixian Bridge—Zijiang River in Jiangnan Town (Liangzuo Tea House, Dehe Tea House, Liangjia Pier in Jiangnan, Wufugong Pier). As the most important section of Anhua's ancient tea route, it witnessed the historical process of the prosperity and decline of the tea trade in the ancient Meishan area. At the same time, this section of the ancient tea road involves a variety of landscape types and there are great differences in visual beauty between different scenes, mainly natural landscape, rural landscape, or town landscape. Because the historical relics need to be kept intact and the modernization of the surrounding environment varies, there are also large visual beauty differences between different viewpoints within the same scene. The above two aspects of the difference in characteristics just met the needs of the research questions in this study. In addition, the eight scenes in Anhua County were not collected at the Yaozijian Ancient Road but were selected as modern townscapes within the radiation of tourism support services along the ancient tea horse road, forming a strong contrast in style with other scenes, which is more capable of supporting the issues explored in this paper. The remaining 53 scenes were all located on the Yaozijian Ancient Road (as shown in Figure 1).

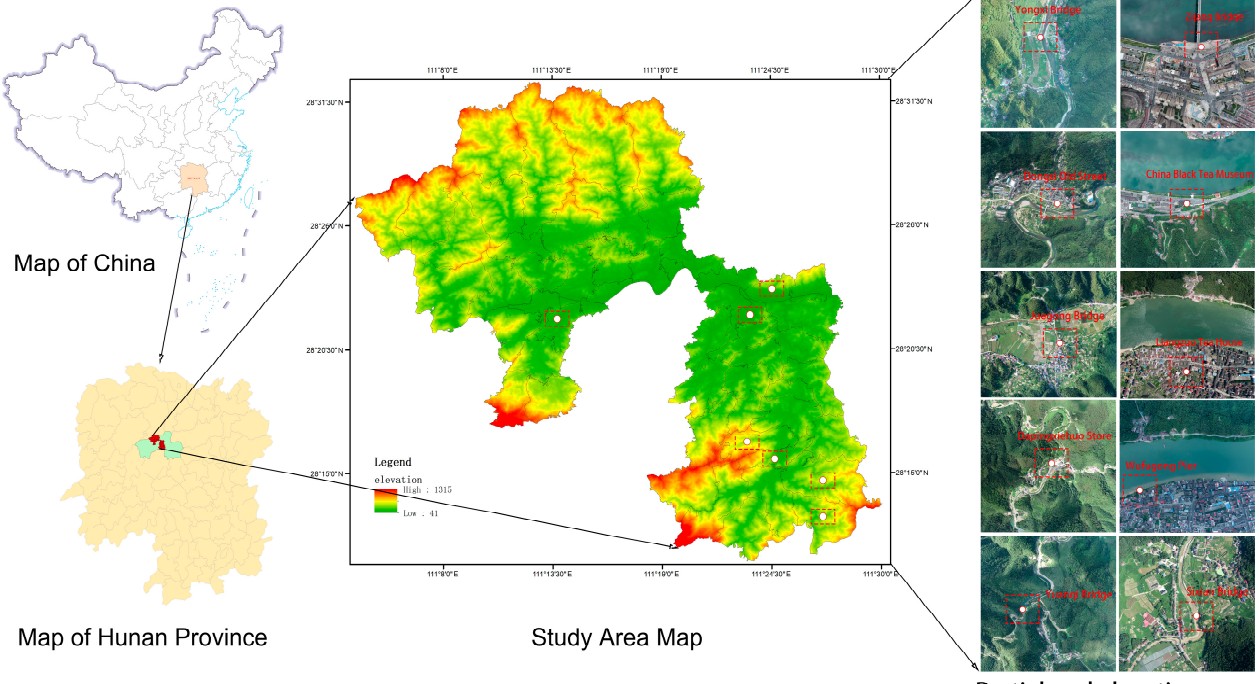

**Figure 1.** Study area.

### 3.2. Site Photos and VR Scenes

Sixty-one panoramic photographs were taken of selected representative landscape nodes within the study area. These photos were taken between 08:00 and 18:00 in clear weather in April 2019 and September 2021, respectively, and were able to reproduce the landscape elements within the scenes. To ensure that the images were captured under relatively uniform environmental conditions, a series of shooting guidelines were developed

for this study: 1. Photographs were taken in sufficient natural light without the use of flash; 2. Photographs were taken using a panoramic camera to maximize the reflection of the landscape within the field of view; and 3. The photographer remained upright with the lens at the same height as both eyes to ensure a consistent angle.

In the next steps, we invited nine university professors majoring in landscape design and visual design as experts to evaluate these 61 photos, from which we selected 20 panorama photos of urban and rural scenes with the most distinctive characteristics of both kinds of differences as the samples for the empirical study. To ensure that the selected photos present as many positive and negative perspective differences as possible, this study sought to take this into full consideration in the selection of experts. At the same time, this study referred to a pre-drafted list of landscape perception evaluation indicators to ensure that the screened landscape samples could reflect the landscape patterns of each survey area as much as possible, and these samples were categorized and labeled. With the premise of minimizing the testers' testing time, we made efforts to present the original appearance of the landscape space in the region comprehensively.

To provide experts with a comprehensive understanding of the site, this study also collected 192 general photographs and 28 video recordings from each site, which demonstrated the various evaluation elements of the site and provided them with additional information.

Through the above methods, a set of high-quality panoramic photographs and related supporting materials were obtained (as shown in Figure 2).

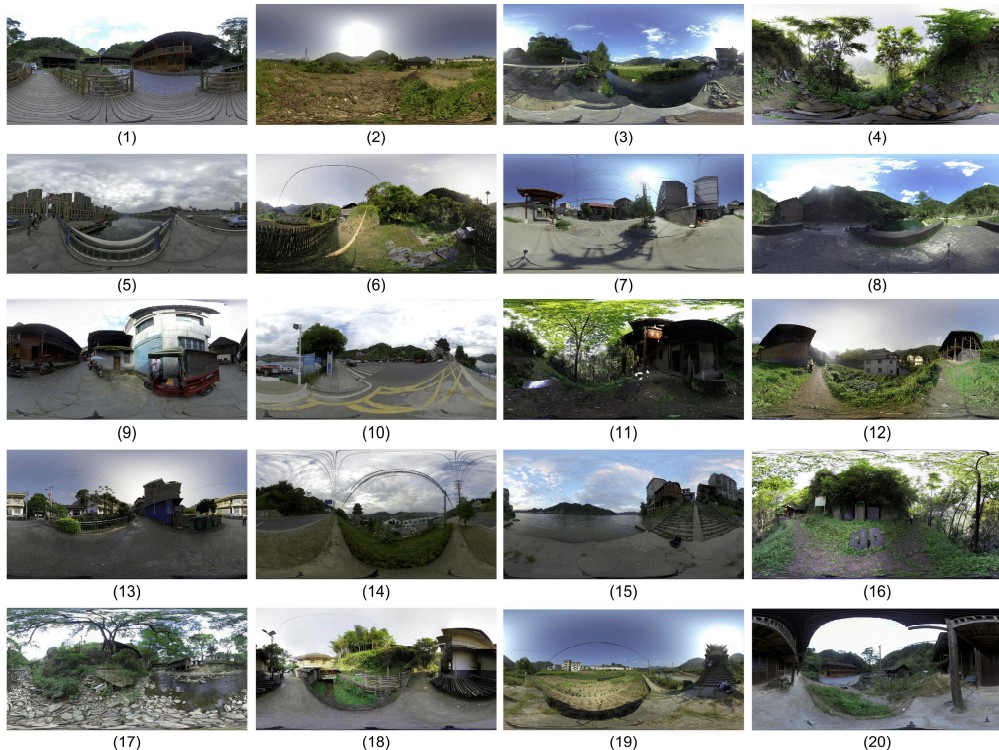

**Figure 2.** Content of filtered panoramic photos. Note: 20 photographs represent 20 different urban and rural landscapes.

### 3.3. Respondents

Studies have shown that the aesthetic scales of evaluators from different cultural backgrounds do not differ statistically significantly and that experts in design and university students have better aesthetic perceptions than the general public [40,41]. In particular, college students, as an urban youth population, have a gradually increasing voice in the family, and their behaviors and preferences have an impact on family travel plans. In addition, young people have high energy and a strong ability to travel, and they are the main group of tourism consumers and the main participants in rural tourism [42–44].

Urban youth are more receptive to new things and master the operation of computers and VR experience software, which helped ensure the smooth conduct of this experiment and to obtain experimental data with complete and reliable references. Therefore, it is hoped that this study will provide strong support for the construction of rural and urban landscapes.

In consideration of shortening the subjects' experimental operation time to ensure that their ratings truly reflect their feelings, this study chose to organize students' participation in the experiment in a school computer room to obtain sample data for subjective evaluation (as shown in Table 1 and Figure 3). Meanwhile, the uniform scores of objective indicators were determined using expert evaluation.

**Table 1.** Experimental organization scheme of landscape perception evaluation cited.

| Category | Number of Persons | Group Composition | Evaluation Type |
|---|---|---|---|
| Expert Group | 9 | University professors and experts in environmental design and visual design | Objective evaluation |
| Student Group | 152 | Undergraduate students of environmental design and other art and design majors in the School of Architecture and Art of Central South University | Subjective evaluation |

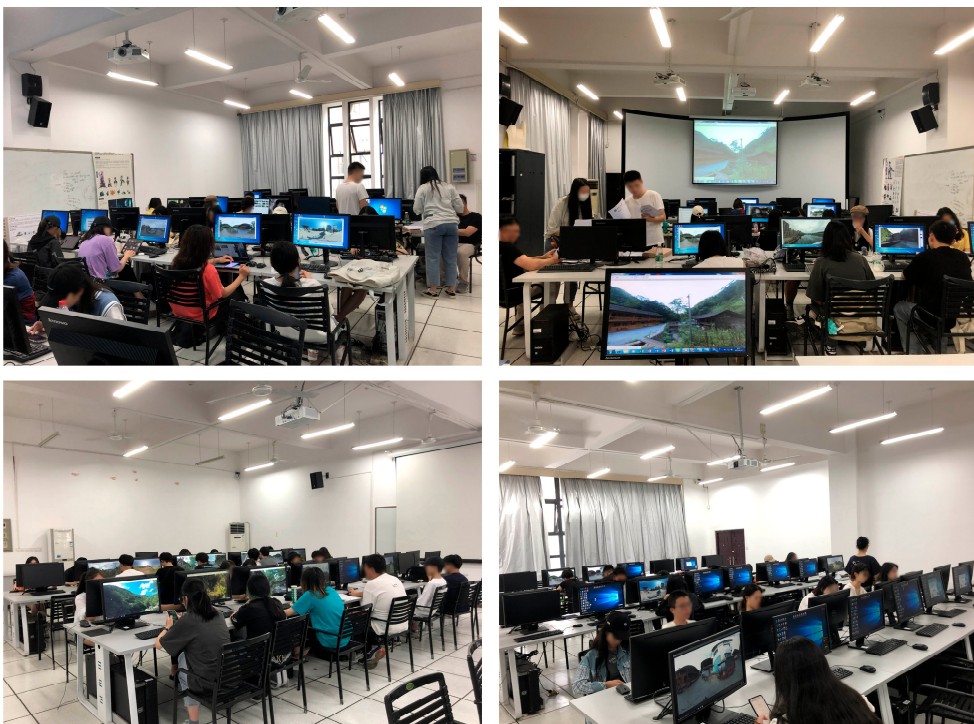

**Figure 3.** Organizing students' experimental evaluation.

*3.4. Experimental Design*

3.4.1. Extraction of Landscape Factors

After sorting out the existing studies in the field of landscape evaluation, a set of landscape perception evaluation indicator systems was initially constructed mainly based on the content of relevant studies published by Rogge E et al. [45–57], and was then adjusted

by combining the characteristics of the sample and the effect of small-scale pre-experiments in this study to ensure that each indicator was reflected in each panorama. At the same time, the subjects conducted scoring experiments to ensure that there were no critical problems such as confusion. A scale of measurement for the village landscape perception evaluation indicators applicable to this study was finally determined (as shown in Table 2 below). The element layer was divided into four major categories: overall environment, sky, soft elements, and man-made elements, of which six indicators were evaluated by the public (C1, C2, C3, C4, C8, C15) and eleven indicators were evaluated by experts (C5, C6, C7, C9, C10, C11, C12, C13, C14, C16, C17), totaling seventeen indicators.

**Table 2.** Experimental organization scheme of landscape perception evaluation cited.

| Variable | Abbreviation | Scoring | | | | |
|---|---|---|---|---|---|---|
| | | 1 | 2 | 3 | 4 | 5 |
| Overall environment | B1 | | | | | |
| Visual harmony of the scenery [38] | C1 | extremely conflicted | more conflicted | moderate | more harmonious | Extremely harmonious |
| Color Beauty [39] | C2 | extremely gray and monotone | more gray and monotone | moderate | more vivid and rich | extremely vivid and rich |
| Historic ambience [40,41] | C3 | extremely thin | relatively thin | moderate | relatively thick | extremely thick |
| Sense of order and regularity | C4 | extremely disorganized | more disorganized | moderate | more regular | extremely regular |
| Visual hierarchy [42] | C5 | extremely thin | relatively thin | moderate | relatively rich | extremely rich |
| Visual area depth | C6 | very short distance | short distance | moderate | long distance | very long distance |
| Dirt and pollution level [40,42,43] | C7 | extremely clean | relatively clean | moderate | relatively dirty | extremely dirty |
| Sky | B2 | | | | | |
| Skyline beauty | C8 | very weak beauty | weak beauty | moderate | strong beauty | very strong beauty |
| Sky ratio [44] | C9 | 0–10% | 10–20% | 20–33% | 33–45% | 45–100% |
| Cloud ratio | C10 | all cloudy | overcast, cloudy | sunny, cloudy | sunny, lightly cloudy | sunny, no clouds |
| Native elements | B3 | | | | | |
| Vegetation ratio [45] | C11 | 0–5% | 5–25% | 25–50% | 50–75% | 75–100% |
| Percentage of exposed soil [46] | C12 | 0–5% | 5–25% | 25–50% | 50–75% | 75–100% |
| Type of vegetation [42,45,47] | C13 | Very few species | relatively few species | moderate | relatively rich in species | extremely rich in species |
| Water ratio [48] | C14 | 0–5% | 5–25% | 25–50% | 50–75% | 75–100% |
| Man-made elements | B4 | | | | | |
| Overall beauty of the man-made object [49] | C15 | extremely ugly | uglier | moderate | more beautiful | extremely beautiful |
| Proportion of man-made objects [39,44,47] | C16 | 0–5% | 5–25% | 25–50% | 50–75% | 75–100% |
| Near-natural degree of flooring material [50] | C17 | 0–5% | 5–25% | 25–50% | 50–75% | 75–100% |

### 3.4.2. Hypothesized Model

In this study, SmartPLS 3.0 was used as the statistical modeling technique. Based on the preliminary theoretical foundation and experimental research, we constructed a structural model for analysis using the scenic beauty degree from different viewpoints as the mediating variable, the overall scenic beauty degree (OSBD) of the scene as the dependent variable, and the landscape factor as the independent variable (as shown in Figure 4). The following research hypotheses were proposed in this study:

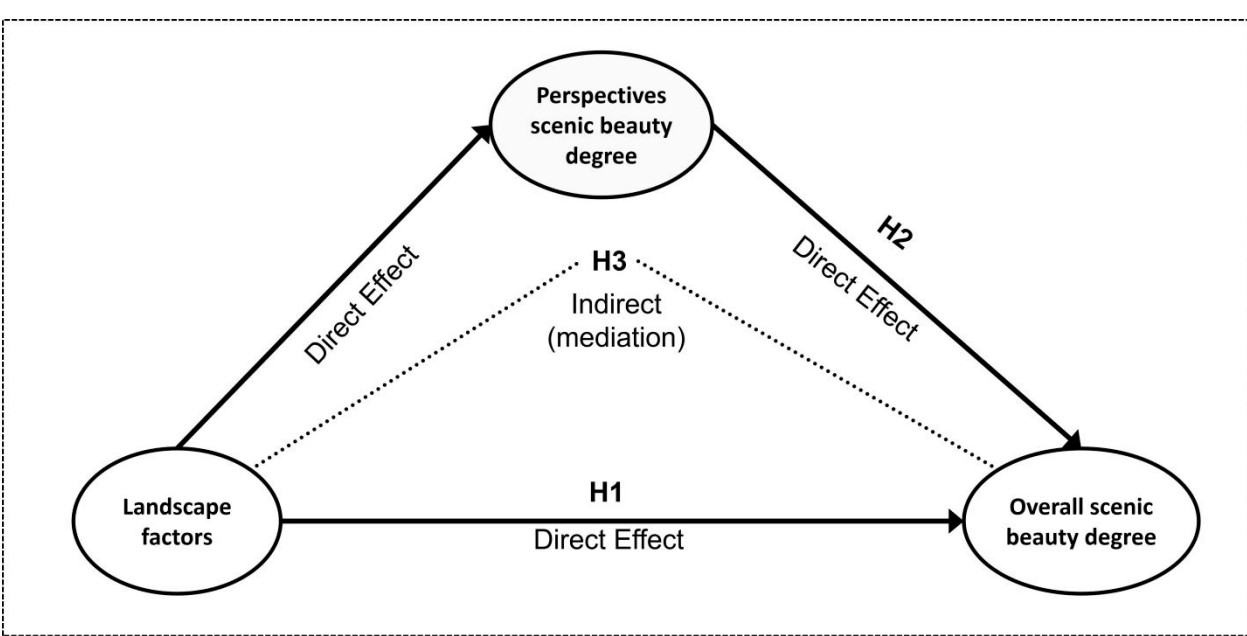

**Figure 4.** Hypothetical model.

**Hypothesis 1.** *There is an influential relationship between landscape indicators and the overall beauty of the scene (H1);*

**Hypothesis 2.** *There is a direct influence of positive/negative viewpoint of scenic beauty on the OSBD (H2);*

**Hypothesis 3.** *There is a significant mediating role of positive/negative viewpoint of scenic beauty in the relationship between landscape indicators and OSBS (H3).*

3.4.3. Measurement of Variables

Compared with the traditional landscape visual evaluation experiments, the subjects in this experiment used the "computer screen + indoor large screen projection" method to view the perceptual 720° VR panorama, which can enable the subjects to perceive the elements in the scene more comprehensively and show them a more comprehensive and realistic scene situation as much as possible, which helps the subject's score to be more similar to that of the scene. First, the 20 scene samples were disordered and renumbered. The organizer first explained the purpose, process, and requirements of the test to the judges about 30 min before the evaluation but did not involve standardized descriptions of the details of the judged subjects.

The first session of the experiment. In scoring the OSBD, the scenic beauty degree evaluation method (SBE) was used. The subjects viewed 20 panoramas by themselves through panorama projection software (DevalVR player) and rated the OSBS of the 20 scenes in order according to their overall impression. The response scale was based on a 5-point scale, with five levels of "dislike, dislike, average, like, and like very much", and the corresponding scores were 1, 2, 3, 4, and 5 in order.

The second part of the experiment. The group extracted the landscape elements reflected in the panorama map and constructed a landscape perception evaluation index system for Hunan Yiyang villages and towns. Based on this, the experimental subjects rated the content corresponding to the evaluation indexes in these 20 scenes in turn according to their feelings when viewing the 20 panoramic pictures. To ensure that the subjects' scoring

could reflect their subjective feelings, the process required the subjects to avoid mutual communication and reference.

The third session of the experiment. Based on the premise of panorama technology and the purpose of the experiment, the experiment added a session on the scenic beauty of positive and negative viewpoints. The expert group cut out 20 scenes of positive perspective (numbered 1–20) and negative perspective (numbered 21–40) photos in advance (as shown in Figure 5). The subjects were asked to set aside their overall impression of the scene and rate the beauty of these 40 photos in turn according to their feelings at the moment.

The fourth session of the experiment. The experts were called into an online meeting and the expert leader showed 20 scenes in turn through screen sharing and divided into 11 rounds to ask opinions on the objective indicator scores of the scenes, with each round scoring a single indicator 20 times, and finally filling in the scoring table with the scores unanimously approved by the nine experts, which was intended to ensure a more objective and fair score for each scene through horizontal comparison.

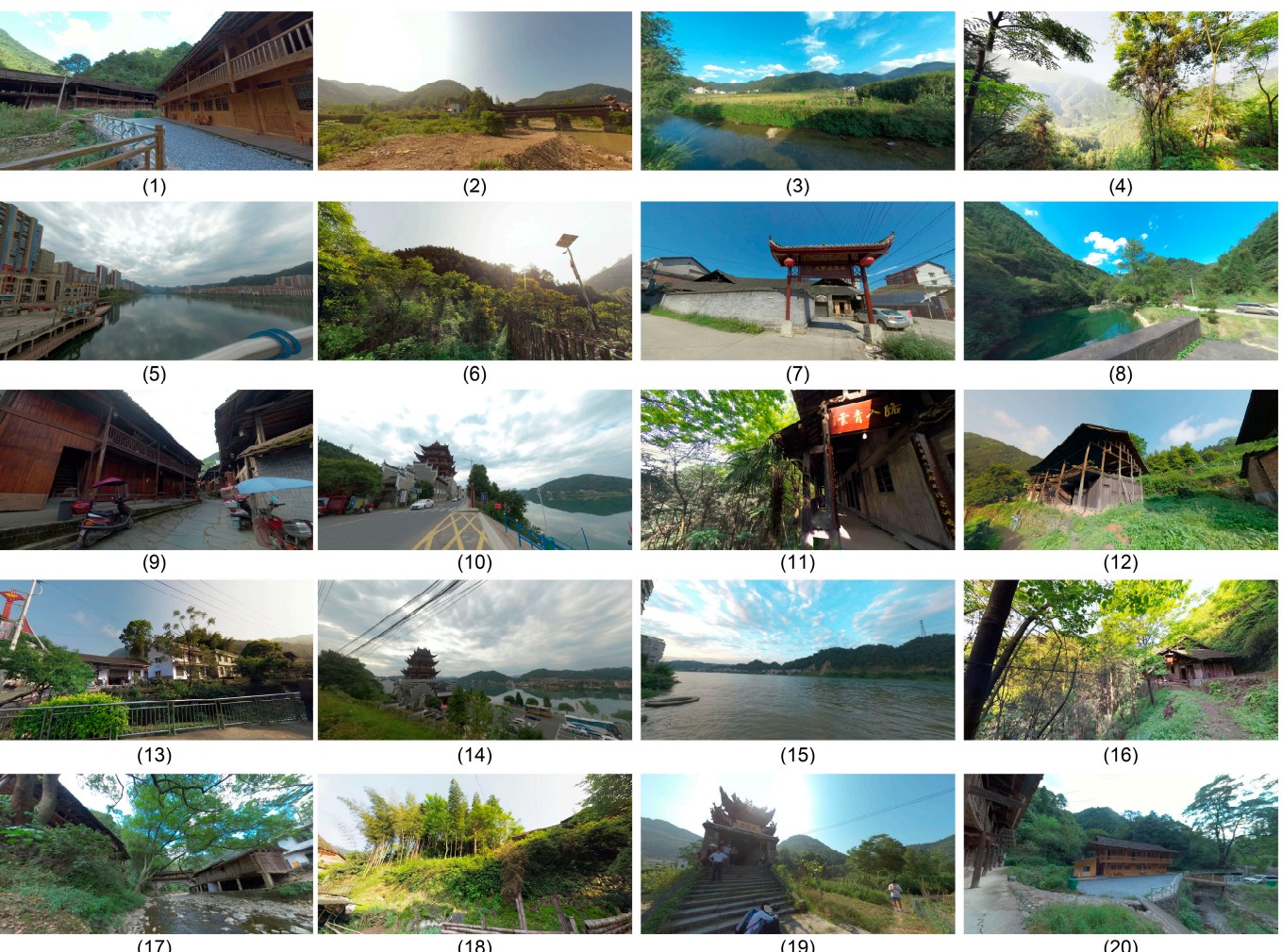

**Figure 5.** *Cont.*

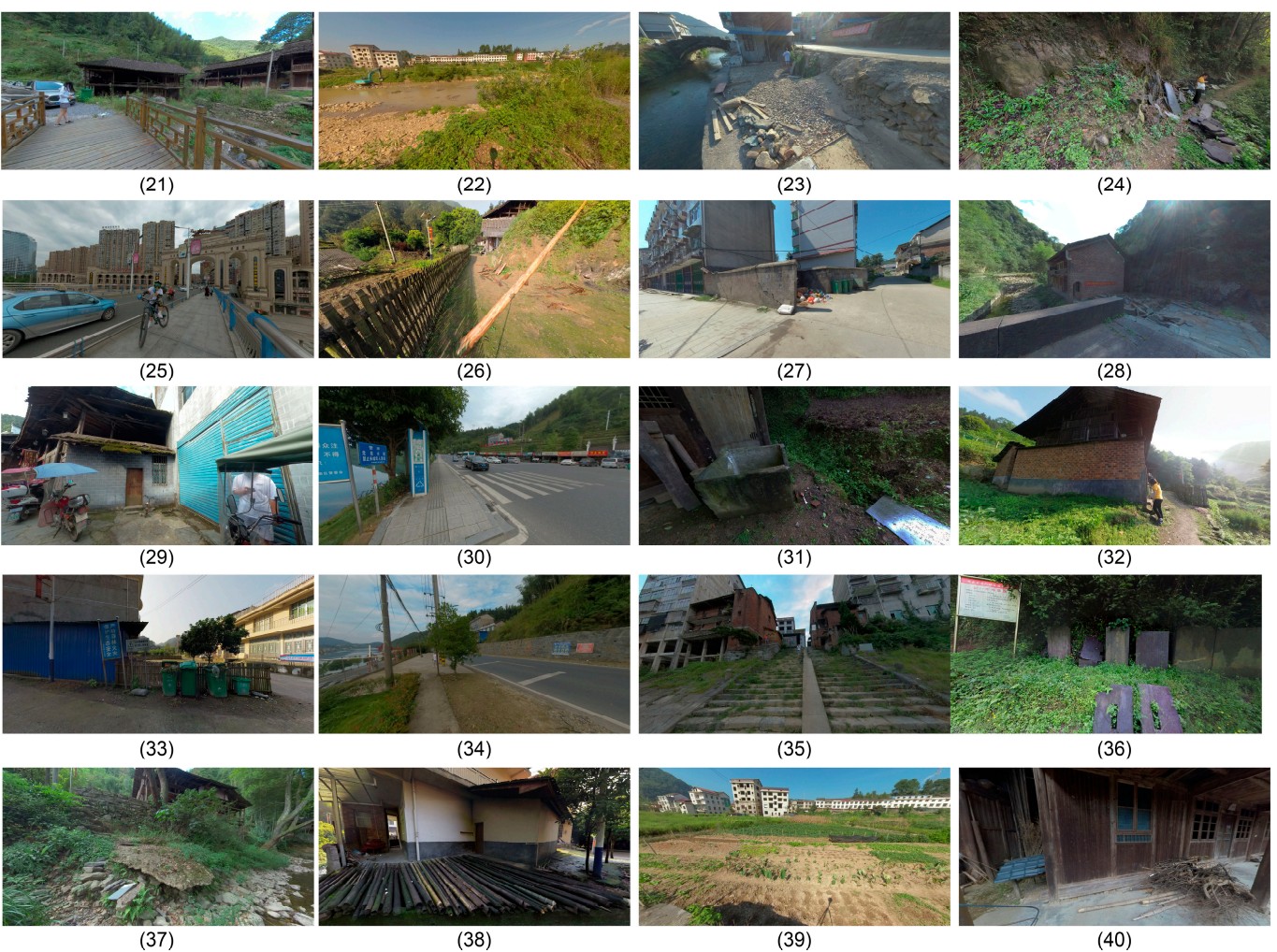

**Figure 5.** Positive view screenshots and negative view screenshots of 20 scenes. Note: Serial numbers 1–20 represent photographs taken from positive viewpoints and 21–40 represent photographs taken from negative viewpoints.

### 3.4.4. Sampling and Sample Size

One-hundred-and-fifty-two experimental subjects and nine experts in this field were recruited as evaluation subjects to conduct subjective evaluations of 20 scenes in two towns and five villages. The experiment obtained 157 samples of subjective evaluation data, eliminated 26 samples participating in the pre-experiment, and finally retained 131 samples, totaling 2620 data. Nine experts conducted an objective evaluation in the form of talks and discussions, and made a unified decision on the objective evaluation indicator scores, obtaining 220 data points.

### 3.4.5. Test Method

1. Reliability test: to ensure the reliability and consistency of the measurement instruments, this study conducted a reliability test on the sample data obtained from the experiment. In the reliability test, internal consistency indicators, such as Cronbach's alpha coefficient, were calculated for each measurement item, and the correlation between the measurement items and the total score was assessed.
2. Validity test: To verify the validity and accuracy of the measurement instrument, a validity test was conducted in this study. In the validity test, the degree of correlation between the measurement items and the theoretical concepts was assessed using the construct validity and reflective validity methods.

3. Covariance test: To exclude a high correlation between independent variables, this study conducted a covariance test. The degree of covariance between the independent variables was assessed by calculating the variance inflation factor (VIF) and the number of conditions, and variables that could lead to multiple covariances were excluded.

4. Correlation test: To determine the correlation between the variables, this study conducted a correlation test. The correlation between the independent and dependent variables was assessed by calculating the Pearson correlation coefficient.

5. Mediating effects test: In SmartPLS 3.0, this study first constructed a structural equation model based on the research objectives and theoretical assumptions. This included identifying the independent, mediating, and dependent variables and defining the relationships among them; second, in evaluating the structural model, this study used path analysis to test the direct effects of the independent variables on the dependent variable. By analyzing the path coefficients, t-values, significance levels, and R-squared values, it was possible to determine the degree of influence and statistical significance of the independent variables; finally, to test the mediating effects, the Bootstrap method was used to calculate the confidence intervals and significance of the indirect effects.

## 4. Results

### 4.1. Reliability and Validity Tests

Based on the sample data obtained from this experiment, a reliability test was conducted and the Cronbach's alpha coefficient was 0.884, indicating that there is internal consistency among multiple items of each variable and the reliability of the sample data is good. At the same time, the validity of the sample data was tested, and the KMO value was 0.911, and the *p*-value was 0.000 (significant), which indicated that the variables were suitable for factor analysis and the index system had good coverage and scientific validity.

### 4.2. Covariance, Correlation Test

First, this study examined the correlations among the variables, as well as assessed whether covariance among the predictor variables would cause problems in the structural model. In this study, after correlation analysis of the sample data, it was found that only the C12 indicator was not significantly correlated with the scenic beauty and was not necessary to exist in the subsequent model calculation, and the C12 indicator (percentage of exposed soil) was excluded (as shown in Figure 6). After covariance statistics, it was found that the VIF values of the C6 and C11 indicator scores were greater than 10 and, to ensure the best model construction, the C6 and C11 indicators (visual area depth, vegetation ratio). Finally, the remaining 14 indicators were tested for covariance again, in which the VIF values were all greater than 0.20 and less than 5.00, indicating that the model did not have multiple covariance problems, the model was well constructed, and the variables had significant correlations within a reasonable range, and the structural model construction and evaluation could be continued in the next step (as shown in Table 3).

**Table 3.** Test results of the passed multicollinearity problem.

| Indicator Code | Beta | t | Tolerances | VIF |
|---|---|---|---|---|
| C1 | 0.174 | 7.512 | 0.393 | 2.546 |
| C2 | 0.139 | 6.021 | 0.398 | 2.516 |
| C3 | 0.116 | 5.649 | 0.502 | 1.992 |
| C4 | 0.049 | 2.321 | 0.477 | 2.096 |
| C8 | 0.061 | 3.07 | 0.544 | 1.838 |
| C15 | 0.108 | 5.113 | 0.472 | 2.117 |
| C5 | 0.009 | 0.352 | 0.298 | 3.359 |
| C7 | −0.104 | −4.402 | 0.378 | 2.648 |
| C9 | −0.144 | −5.816 | 0.346 | 2.887 |
| C10 | 0.032 | 0.988 | 0.204 | 4.899 |

**Table 3.** *Cont.*

| Indicator Code | Beta | t | Tolerances | VIF |
| --- | --- | --- | --- | --- |
| C13 | 0.008 | 0.333 | 0.362 | 2.76 |
| C14 | 0.09 | 3.036 | 0.239 | 4.19 |
| C16 | −0.004 | −0.167 | 0.456 | 2.191 |
| C17 | −0.027 | −1.348 | 0.515 | 1.944 |

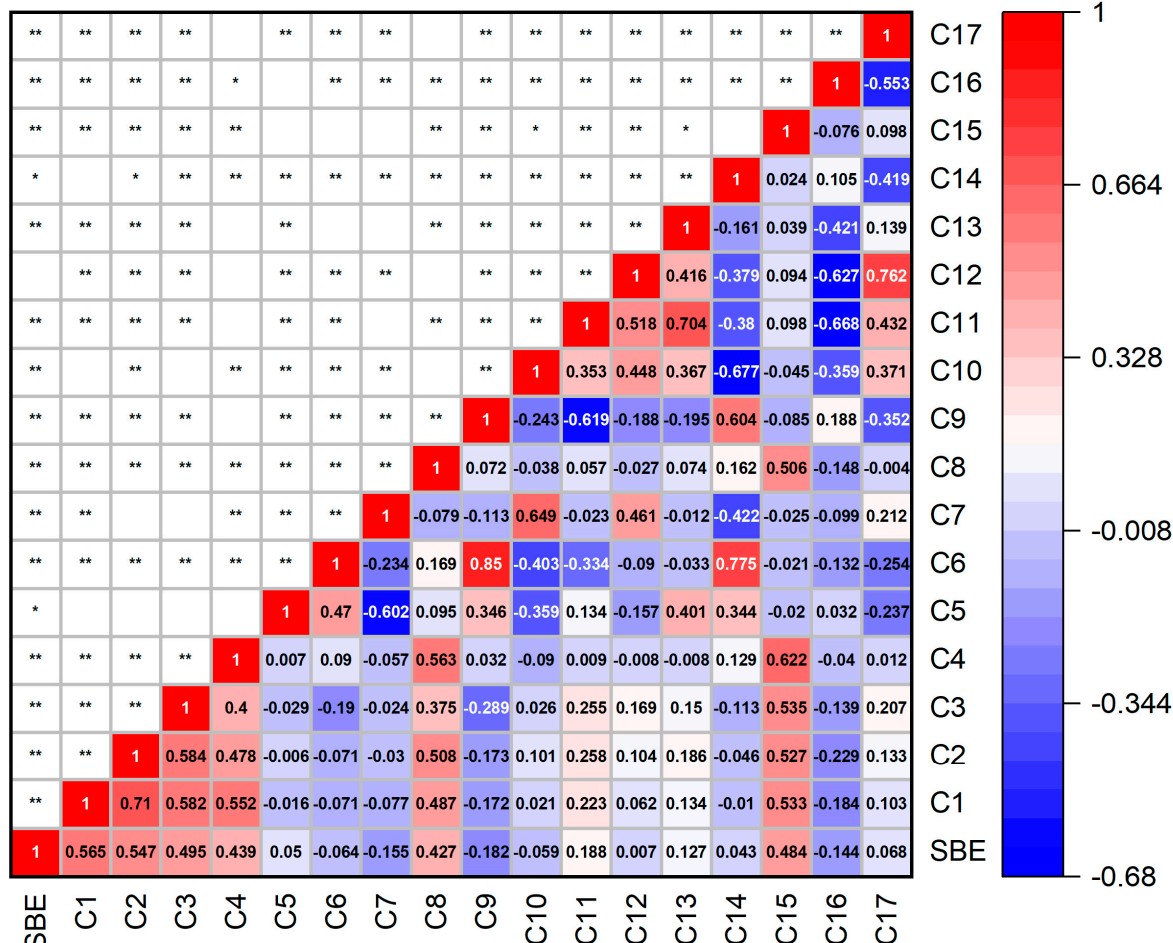

**Figure 6.** Correlation heat map. Note: Significance levels: "**" $\leq$ 0.001, "*" $\leq$ 0.05.

### 4.3. Meditational Model Evaluation

The $R^2$ test was conducted in this study and the $R^2$ value was 0.45, indicating that the model possesses good predictive accuracy. In the mediation model, it was first tested whether the direct relationship between landscape indicators and scenic beauty was moderated by the mediation effect of positive/negative viewpoint scenic beauty. The structural equation model performs the mediation effect analysis in several steps: in the first stage, it assesses the direct effect, i.e., the direct effect of landscape indicators on scene beauty perception (OSBSP), and the direct effect of positive/negative viewpoint of scenic beauty perception on OSBSP, while excluding the mediation effect between both landscape indicators and OSBSP. As the data in Table 4 show, the path coefficients between the nine landscape indicators C1, C2, C3, C4, C7, C8, C9, C14, and C15 and OSBSP are significant, indicating that the indicator variables consisting of 14 indicators have an extremely strong direct effect on OSBSP, supporting research hypothesis H1. The path coefficients between positive and negative viewpoints of scenic beauty perception and OSBSP are significant, and the path coefficients of positive viewpoint scenic beauty perceptions (PVSBP) were

greater than negative viewpoint scenic beauty perceptions (NVSBP), indicating that the direct influence of PVSBP on OSBSP was greater, thus supporting research hypothesis H2.

**Table 4.** Direct effect results affecting OSBS.

| Indicator Code | PVSBD | NVSBD | OSBD (without Mediation) |
|---|---|---|---|
| C1 | 0.144 *** | 0.054 ** | 0.174 *** |
| C2 | 0.17 *** | 0.061 ** | 0.139 *** |
| C3 | 0.02 | 0.123 *** | 0.116 *** |
| C4 | 0.1 *** | 0.052 ** | 0.049 ** |
| C5 | 0.115 *** | −0.113 *** | 0.009 |
| C7 | −0.003 | −0.025 | −0.104 *** |
| C8 | 0.133 *** | 0.103 *** | 0.061 ** |
| C9 | −0.157 *** | 0.077 ** | −0.144 *** |
| C10 | 0.12 *** | −0.036 | 0.032 |
| C13 | −0.109 *** | 0.072 ** | 0.008 |
| C14 | 0.112 *** | 0.076 ** | 0.09 ** |
| C15 | 0.087 *** | 0.162 *** | 0.108 *** |
| C16 | 0.128 *** | 0.015 | −0.004 |
| C17 | 0.012 | 0.051 ** | −0.027 |
| PVSBD | | | 0.122 *** |
| NVSBD | | | 0.062 *** |

Note: Significance levels: *** $\rho \leq 0.001$, ** $\rho \leq 0.05$. OSBD means SBE value of the scene; PVSBD means SBE value of the positive viewpoint; NVSBD means SBE value of the negative viewpoint.

The next step was to test for mediating effects. This study utilized the variance as a percentage (VAF) calculation to determine the extent to which mediating variables absorb the direct relationship [58–60]. It determines the extent to which the variance in OSBSP is explained by the indirect effect of positive/negative viewpoint scenic beauty perception. As shown in Table 5, C5 (Visual hierarchy), C10 (Cloud ratio), C13 (Type of vegetation), and C16 (Proportion of man-made objects) significantly mediated the OSBS through the PVSB with VAF of 46.7%, 30.6%, 52.0%, and 76.2%, respectively. In addition, C5 (visual hierarchy of the landscape) had a significant mediating effect on scenic beauty through negative perception, with a VAF of 23.3%. The above results are all "partially mediated" effects and therefore support the research hypothesis H3.

**Table 5.** Mediating effect results affecting OSBS.

| IS to OSBD (with Mediation) | Indirect Effect | | Total Effect | VAF | |
|---|---|---|---|---|---|
| | Mediator Variable | | | Mediator Variable | |
| | PVSBD | NVSBD | | PVSBD | NVSBD |
| C1 | 0.018 *** | 0.003 * | 0.195 *** | 9.2% | 1.5% |
| C2 | 0.021 *** | 0.004 * | 0.164 *** | 12.8% | 2.4% |
| C3 | 0.002 | 0.008 ** | 0.126 ** | 1.6% | 6.3% |
| C4 | 0.012 *** | 0.003 * | 0.064 *** | 18.8% | 4.7% |
| C5 | 0.014 *** | −0.007 ** | 0.03 | 46.7% | 23.3% |
| C7 | 0.001 | −0.002 | 0.106 | 0.0% | 1.9% |
| C8 | 0.016 *** | 0.006 ** | 0.083 *** | 19.3% | 7.2% |
| C9 | −0.019 *** | 0.005 ** | 0.168 ** | 11.3% | 3.0% |
| C10 | 0.015 ** | −0.002 | 0.049 ** | 30.6% | 4.1% |
| C13 | −0.013 *** | 0.004 ** | 0.025 ** | 52.0% | 16.0% |
| C14 | 0.014 ** | 0.005 * | 0.109 *** | 12.8% | 4.6% |
| C15 | 0.011 ** | 0.01 *** | 0.129 *** | 8.5% | 7.8% |
| C16 | −0.016 *** | 0.001 | 0.021 *** | 76.2% | 4.8% |
| C17 | 0.001 | 0.003 * | 0.031 | 3.2% | 9.7% |

Note: Significance levels: *** $\rho \leq 0.001$, ** $\rho \leq 0.05$, * $\rho \leq 0.1$. OSBD means SBE value of the scene; PVSBD means SBE value of the positive viewpoint; NVSBD means SBE value of the negative viewpoint. When VAF < 20%, there is no mediation effect; when 20% < VAF < 80%, it is judged as a "partial mediation" effect, and when VAF > 80%, it is judged as a "full mediation" effect.

## 5. Discussion

### 5.1. Landscape Factors that Have a Direct and Significant Impact on the OSBS

As shown in Figure 7, the results of this study indicate that C1 (visual harmony of the scenery), C2 (color Beauty), C3 (historic ambiance), C4 (sense of order and regularity), C7 (dirt and pollution level), C8 (skyline beauty), C9 (sky ratio), C14 (water ratio), and C15 (proportion of man-made objects) were all found to play an important role in the observer's beauty experience.

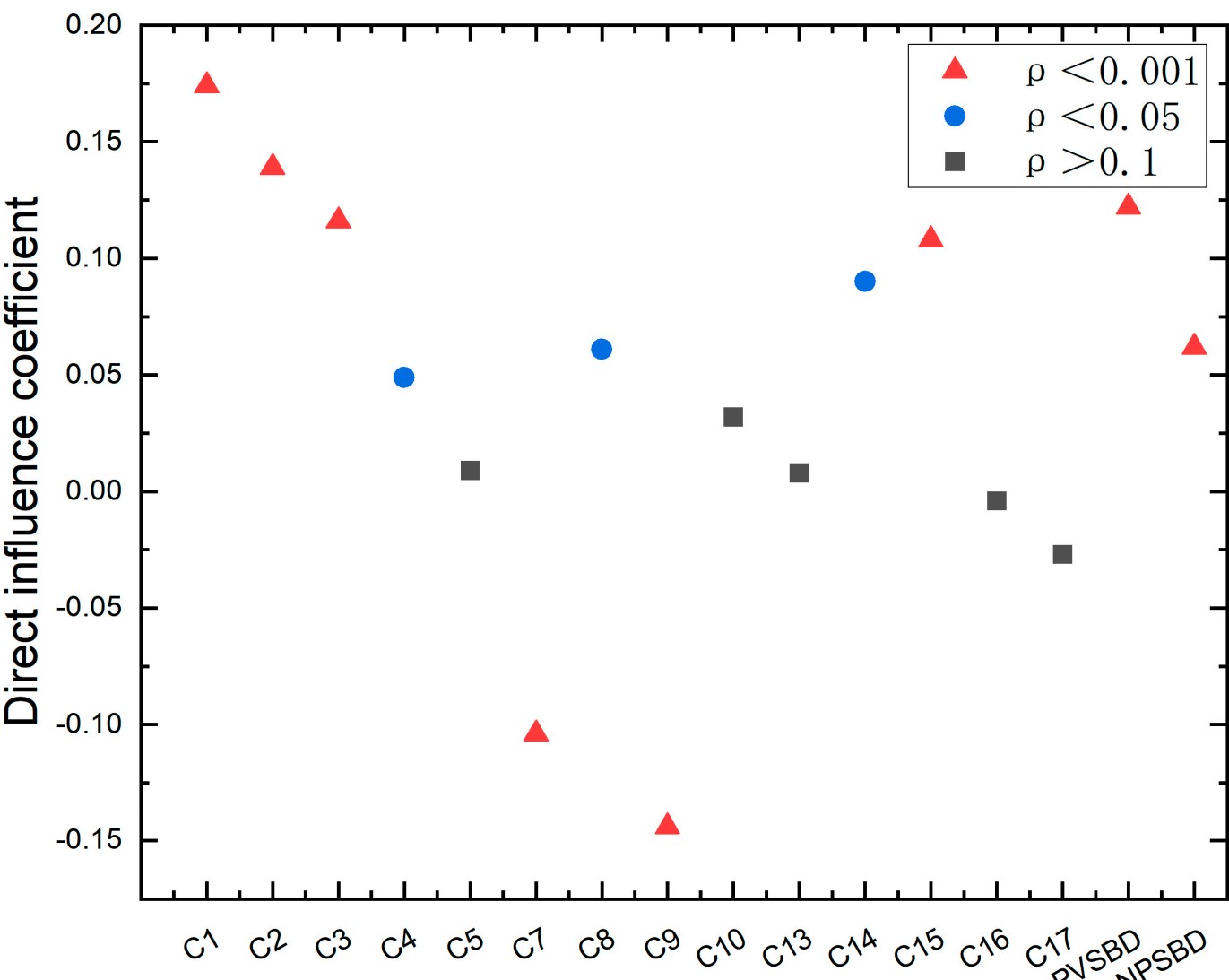

**Figure 7.** Direct effect results affecting the OSBS.

Firstly, C1 positively affects the evaluation of the OSBS, as it imparts a sense of balance and harmony to the panorama, aligning with people's beauty preferences. This result is consistent with previous studies and supports the importance of C1 on the OSBS [4,61–63]. Secondly, C2 also plays a key role in the evaluation of the OSBS. Brighter, moderately saturated, and color-coordinated scenes tend to be more popular. Therefore, the presence of C2 can enhance the observer's beauty experience of the panorama [49,64–67]. Moreover, the presentation of C3 has a significant positive impact on the beauty evaluation. Scenes conveying historical, cultural, and traditional elements evoke emotional resonance in observers, thereby enhancing their beauty evaluation. This finding aligns with social research highlighting the influence of historical and cultural factors in aesthetic perception [68–70]. Furthermore, C4 exhibits a positive association with the evaluation of OSBS. Observers

are more inclined to have a positive beauty experience of an ordered and neat scene. This may be related to human preferences for clarity, legibility, and simplicity [71]; On the other hand, C7 negatively influences the evaluation of OSBS. Observers tend to perceive clean and tidy scenes as beautiful and appealing [72–75]. Consequently, a higher level of dirtiness and pollution in the panorama can negatively impact observers' beauty experiences. C8 has a positive effect on the OSBS evaluation. This shows that the shape, height, and curved features of the skyline have an impact on the viewer's beauty experience. When the skyline in the panorama has a unique and beautiful shape, appropriate height, and smooth lines, observers are more likely to have a positive aesthetic experience. C9 has an important negative impact on scenic beauty. A disproportionately large sky area can result in a scene lacking focus and direction for the viewer's eye. Landscape elements are often employed in design to guide the viewer's gaze and eye flow, creating a compelling focal point. Nevertheless, an excessive sky area can dilute the overall focus, leading to a diminished beauty experience. Further investigation is needed to pinpoint the exact reasons for this effect. C14 plays an important positive role in the OSBS evaluation. The viewer is more likely to have a positive beauty experience when a moderate amount of water is present in the scene and the characteristics of the water body can be reasonably displayed. This result is consistent with previous studies that found that the presence of water bodies can increase the visual appeal and beauty experience of a scene [76,77]. The reflection and flow of water bodies can bring dynamism and variability, thus enhancing the vividness and fascination of the scene. Finally, C15 also shows a significant positive effect on the beauty evaluation. Viewers are more likely to perceive artifacts as attractive and aesthetically pleasing scenes when they exhibit characteristics that are unique, exquisite, and in harmony with the environment, which is consistent with findings in architectural beauty regarding the influence of architectural structures on beauty evaluation [78–80]. In conclusion, these findings are in line with existing research on visual perception, beauty preferences, and sociocultural factors, and providing valuable insight and understanding for the field of landscape aesthetics and design.

## 5.2. The Beauty of Viewpoint Has a Direct and Significant Impact on the OSBS

In this study, it was found that positive and negative viewpoints of scenic beauty perceptions have a direct and significant positive impact on the OSBS perceptions, with PVSBP having a greater degree of impact than negative viewpoints. This finding has important theoretical and practical implications in the study of landscape perception evaluation. First, the results indicate that viewers are more inclined to make positive beauty evaluations of positive viewpoints when perceiving scenes. This is consistent with relevant findings in affective psychology, where affective evaluations have an important influence on the experience of beauty. Viewers may be more likely to experience pleasurable and satisfying emotions from the positive viewpoint, which in turn enhances the beauty perception of the whole scene [1,81,82]. In addition, studies have revealed the role of visual guidance and focus creation in the beauty experience. A positive viewpoint is usually effective in guiding the viewer's gaze and eye flow, creating a compelling focal point. This focused composition allows the viewer to immerse themselves more deeply, resulting in a more profound and comprehensive beauty experience [3,16,19]. In contrast, a negative viewpoint may not effectively direct the line of sight, resulting in a distraction of the viewer's attention, thus reducing the focus of the beauty experience. In addition, the findings have been correlated with the association of individual preferences and cultural background. According to existing studies, viewers' individual preferences and cultural backgrounds also have an impact on the beauty experience. Some studies have shown that people are more inclined to make positive comments about positive, beautiful views of the landscape. This is related to individual beauty preferences and cultural values [9–11,83]. Thus, the positive viewpoint of scenic beauty degree (PVSBD) may be supported to some extent by individual preferences and cultural contexts, thus having a greater impact on the OSBS.

In light of the foregoing, the diverse effects of positive and negative viewpoints on the overall aesthetic perception of a scene can be ascribed to an intricate interplay of emotional experiences, heightened aesthetic appreciation, subjective evaluations, and varying points of focus. Specifically, positive viewpoints tend to evoke positive emotional experiences, encompassing joy, satisfaction, and a sense of happiness. Consequently, individuals are more inclined to perceive the landscape as aesthetically pleasing, deeming it delightful and deserving of admiration. As viewpoints and emotions are inherently subjective, positive perspectives may engender favorable evaluations of the same landscape, while contrasting negative viewpoints may result in less favorable assessments. Moreover, positive and negative viewpoints can distinctly shape individuals' focus during the process of perceiving a landscape. Positive viewpoints incline individuals to pay heightened attention to the landscape's beauty and merits, while downplaying or diminishing their focus on any existing flaws. In contrast, negative viewpoints tend to drive individuals to concentrate more on the landscape's problems and shortcomings, thereby overlooking or reducing their attention to its exquisite aspects. In essence, the dichotomy between positive and negative viewpoints introduces intriguing implications for the holistic aesthetic perception of landscapes, shedding light on the multifaceted factors that influence our subjective appreciation of the natural environment. These insights have significant ramifications for landscape design, planning, and management, paving the way for a more comprehensive understanding of the nuanced relationship between human perception and landscape aesthetics.

### 5.3. Landscape Factors That Indirectly Affect the OSBS through the Viewpoint Scenic Beauty

As shown in Figure 8, the results of this study show that C5 (visual hierarchy) plays an important mediating role in the influence of positive and negative viewpoints of scenic beauty on the OSBS, a finding that is supported by existing studies. First, PVSB focuses more on the positive features and beautiful elements of the scene. In this case, the role of visual hierarchy in enhancing the beauty experience may be more obvious. A higher visual hierarchy provides more visual variation and hierarchy, increasing the visual appeal and emotional evaluation of the viewer, which further enhances their evaluation of the OSBS [49]. In contrast, the NVSBP focuses more on the negative features and flaws of the scene. When viewers hold a negative viewpoint, they are more inclined to perceive and focus on the discord, incongruity, and unpleasantness of the scene. In this case, the effect of C5 on the beauty experience may be relatively small. Even though visual hierarchy is higher, it may not change the viewer's negative affective evaluation of the scene as a whole, which is consistent with previous findings, such as a study that found that, in urban landscapes, low-level landscape organization may lead to a sense of chaos and dissonance, triggering negative affective evaluations and thus reducing OSBS perception [64,84]. In addition, individual psychological preferences and cognitive biases may also differ in their mediating influence on positive and negative viewpoint scenic beauty perceptions. Different viewers may differ in the importance of beauty elements and the way they perceive them, which may lead them to perceive and evaluate the visual hierarchy differently and thus influence the extent of its mediating effect [8,11,81,83].

Secondly, C10 (cloud ratio) had a significant positive impact on the OSBS through the PVSBD. This finding can be supported by existing studies. Previous research has demonstrated that the positive viewpoint offers an open and wide visual experience, which is positively correlated with the beauty rating of the scene. Positive viewpoint screenshots typically present expansive views, bright environments, and a positive and pleasant atmosphere, characteristics that are strongly associated with aesthetic perception and satisfaction [4,85,86]. In addition, C10, an important element in the landscape, has a significant impact on the beauty of the scene. A higher cloud ratio usually implies greater sky coverage, creating a light and serene feeling that aligns the sense of openness and positive emotions brought about by a positive viewpoint, thus positively influencing the OSBS [87].

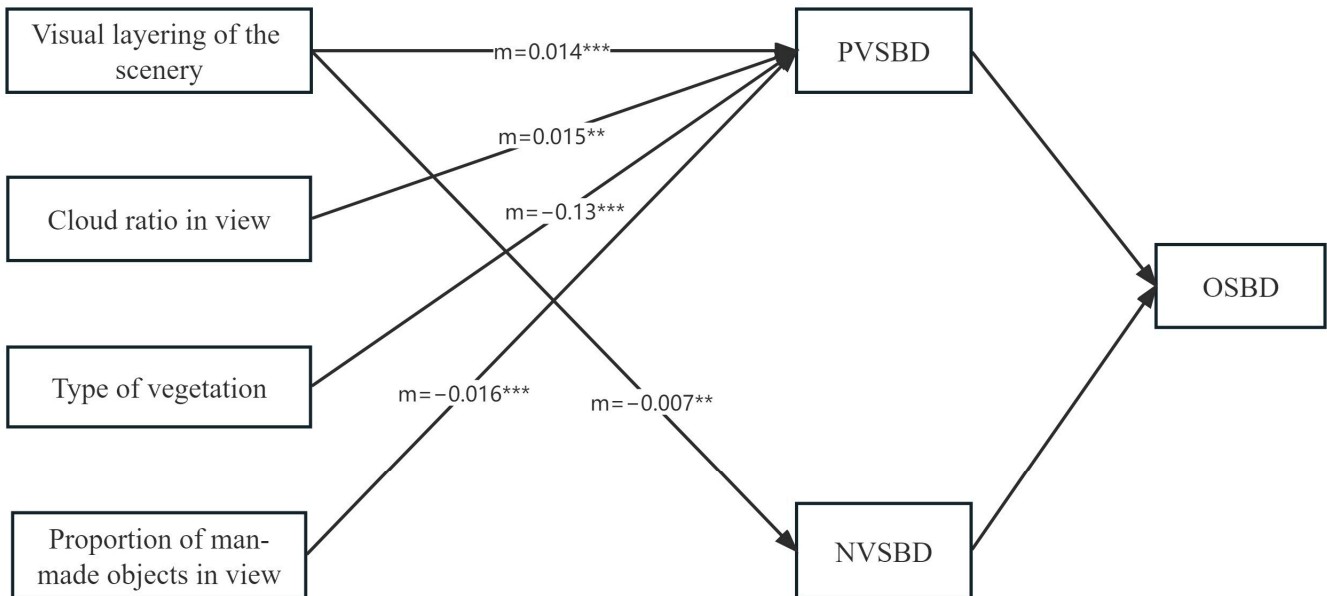

**Figure 8.** Mediating effect results affecting the OSBS. Note: Significance levels: *** $\rho \leq 0.001$, ** $\rho \leq 0.05$. m means indirect effect coefficient. OSBD means SBE value of the scene; PVSBD means SBE value of the positive viewpoint; NVSBD means SBE value of the negative viewpoint.

Furthermore, this study also found that C13 (type of vegetation) had a significant negative effect on the OSBS through PVSBP. This may be attributed to the different perceptual mechanisms and viewing perspectives. The positive viewpoint usually emphasizes the openness and expansiveness of the scene, which may create a visual conflict with the large range of type of vegetation. An abundance of vegetation may make the scene appear too dense or complex, thus reducing the OSBD. Moreover, dense vegetation may also cause visual confusion, making it difficult to focus on the overall scene and consequently impacting the perception of beauty.

In addition, C16 (proportion of man-made objects) was also found to have a significantly negative effect on the OSBS through the PVSBD. One possible explanation is the balance between the naturalness and the artificiality of the landscape. In natural landscapes, man-made objects are often considered to be a disruptive factor in the harmony of nature. Viewers may prefer to look for natural elements in a natural landscape, and excessive or bulky man-made objects may upset this balance [6,79]. The negative impact on the PVSBP may be due to the fact that excessive or bulky man-made objects cause differences in viewers' perceptions of the overall naturalness of the scene, thus reducing the beauty evaluation.

### 5.4. Bias

It is essential to acknowledge that the findings of this dissertation might be affected by sampling bias and constrained by the scope and characteristics of the study sample, which could limit the generalizability of the results. The determinants of scene aesthetics can vary significantly from person to person and are influenced by numerous factors, such as culture, social context, individual experiences, and subjective preferences. As a result, other studies may arrive at different conclusions. To enhance the reliability and generalizability of the findings, future studies should consider employing larger sample sizes and collecting data from diverse regions. This approach would allow for a more comprehensive understanding of the factors influencing scene aesthetics. Additionally, further investigations are necessary to explore the mechanisms through which positive and negative perspectives impact the OSBS in various contexts and among different individuals.

## 6. Conclusions

In this study, 20 different spatial scenes of Anhua County Yaozijian Ancient Road in town and countryside were selected as empirical objects, and 20 photos of positive/negative viewpoints were intercepted to establish a landscape perception evaluation model and analyze the functional relationship between landscape factors, positive and negative viewpoints, and the OSBD of the scenes. As a result, the following important conclusions were drawn:

1. Visual harmony of the scenery, color beauty, historic ambiance, sense of order and regularity, dirt and pollution level, skyline beauty, sky ratio, water ratio, and proportion of man-made objects have a direct and significant impact on the OSBS of the panorama shot. Therefore, in landscape design and planning, due consideration should be given to the handling and presentation of these landscape elements to enhance aesthetic outcomes. Designers can elevate the landscape aesthetics by accentuating these crucial elements, thus crafting more appealing, beautiful, and inviting outdoor spaces.

2. The results of the empirical case study indicate that the overall effect of the positive viewpoint on the OSBS is greater than that of the negative viewpoint. Therefore, in landscape design, emphasizing positive viewpoints and employing strategies to enhance landscape aesthetics can yield superior results, augmenting the allure and affinity of the landscape.

3. Visual hierarchy, cloud ratio, type of vegetation, and proportion of man-made objects play a key role in mediating the influence of PVSB on the OSBS, and their influence is greater. Visual hierarchy reflects the richness and hierarchy of elements in the scene, which in turn has a positive influence on the overall beauty of the viewer. Visual hierarchy reflects the richness and hierarchy of elements in the scene, which in turn positively influences the viewer's OSBS. In addition, visual hierarchy also mediates the OSBS through the negative viewpoint of scenic beauty, although the degree of its mediating influence is relatively low. In landscape design and planning, it is essential to pay attention to and effectively leverage the role of these intermediary factors to enhance the quality and overall aesthetic appeal of the landscape.

These findings have important implications for landscape beauty and scene perception evaluation and provide a scientific basis for landscape design, urban planning, and environmental management. In future practice, we should place emphasis on the treatment and presentation of landscape elements, prioritize the cultivation of positive viewpoints, and effectively utilize the role of intermediary factors. By doing so, we can create more captivating, beautiful, and public-friendly landscape spaces, fostering sustainable development and improvements in urban ecological environments. This study provided new perspectives and methodologies to further advance research in landscape aesthetics evaluation and landscape planning. Future research can further explore other potential influencing factors and delve into the interrelationships among the factors to further improve the understanding and evaluation methods of OSBS.

**Author Contributions:** Conceptualization, B.L., Y.C. and Q.Z.; methodology, Q.Z., Y.C. and B.L.; software, Y.C. and Q.Z.; validation, B.L., Q.Z. and Y.C.; formal analysis, Y.C., Q.Z. and B.L.; investigation, Y.C., Q.Z. and B.L.; resources, B.L., Y.C. and Q.Z.; data curation, Y.C. and Q.Z.; writing—original draft preparation, Y.C., Q.Z. and B.L.; writing—review and editing, B.L., Q.Z., and Y.C.; visualization, Q.Z. and Y.C.; supervision, B.L.; project administration, B.L.; funding acquisition, B.L.. Y.C. and Q.Z. contributed equally to this manuscript. All authors have read and agreed to the published version of the manuscript.

**Funding:** Humanities and Social Science Fund of Chinese Ministry of Education (Number: 22YJA760034).

**Institutional Review Board Statement:** Not applicable.

**Informed Consent Statement:** Not applicable.

**Data Availability Statement:** Not applicable.

**Conflicts of Interest:** The authors declare no conflict of interest.

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
