# Peer review of "Positive or Negative Viewpoint Determines the Overall Scenic Beauty of a Scene: A Landscape Perception Evaluation Based on a Panoramic View"

_sustainability, doi:10.3390/su151411458_

Round 1
Reviewer 1 Report
The study deals with a very interesting topic, which is currently discussed in the academic community. Examining landscape perception contributes to solve numerous problems in the field of tourism, destination marketing, urban planning and many others. The paper has a logical structure and the authors use correct methods. The content is balanced, relies on relevant sources and provides an interesting insight that can be useful in practice as well as for further academic discussion.
In the introductory section, the authors state that landscape beauty perception is a subject of study in the fields of urban planning, environmental design, tourism planning, and psychology. In such cases, it is necessary to give examples of works that have dealt with such research (e.g. tourism planning: Matlovicova, K and Kormanikova, J. 2014. City Brand-Image Associations Detection. Case Study of Prague, International Multidisciplinary Scientific Conferences on Social Sciences and Arts, SGEM 2014, Psychology And Psychiatry, Sociology And Healthcare, Education, Vol II , pp.139-146; urban planning, environmental design: Mocak, P. et al. 2022: 15-Minute City Concept as a Sustainable Urban Development Alternative: A Brief Outline of Conceptual Frameworks and Slovak Cities as a Case. Folia Geographica 64 (1) , pp.69-89), or destination marketing: Matlovicova, K; Kolesarova, J and Matlovic, R. 2016. Selected Theoretical Aspects Of The Destination Marketing Based On Participation Of Marginalized Communities. 8th International Annual Scientific Conference on Hotel Services, Tourism and Education, pp.128-143).
The above comment in no way diminish the quality of the study. It is a quality study based on a well-developed and original methodology for obtaining and processing relevant data.
I definitely recommend it for publication after minor changes.
Author Response
Dear reviewer:
Thanks very much for taking your time to review this manuscript. We appreciate all your generous comments and suggestions! Please find my itemized responses in below and my revisions in the re-submitted files.
- Response to comment:The study deals with a very interesting topic, which is currently discussed in the academic community. Examining landscape perception contributes to solve numerous problems in the field of tourism, destination marketing, urban planning and many others. The paper has a logical structure and the authors use correct methods. The content is balanced, relies on relevant sources and provides an interesting insight that can be useful in practice as well as for further academic discussion..
Response: Our team is very grateful for your recognition and guidance, and we are blessed to have met you. We hope to continue to receive your guidance and recognition in the future. Thank you very much!!!
- Response to comment:In the introductory section, the authors state that landscape beauty perception is a subject of study in the fields of urban planning, environmental design, tourism planning, and psychology. In such cases, it is necessary to give examples of works that have dealt with such research (e.g. tourism planning: Matlovicova, K and Kormanikova, J. 2014. City Brand-Image Associations Detection. Case Study of Prague, International Multidisciplinary Scientific Conferences on Social Sciences and Arts, SGEM 2014, Psychology And Psychiatry, Sociology And Healthcare, Education, Vol II , pp.139-146; urban planning, environmental design: Mocak, P. et al. 2022: 15-Minute City Concept as a Sustainable Urban Development Alternative: A Brief Outline of Conceptual Frameworks and Slovak Cities as a Case. Folia Geographica 64 (1) , pp.69-89), or destination marketing: Matlovicova, K; Kolesarova, J and Matlovic, R. 2016. Selected Theoretical Aspects Of The Destination Marketing Based On Participation Of Marginalized Communities. 8th International Annual Scientific Conference on Hotel Services, Tourism and Education, pp.128-143).
Response: Thank you for your detailed comments. We attach great importance to this comment. We cite relevant literature in these fields in order to support the theory. Your comments are very helpful and we thank you very much. We would very much like to publish this article with your guidance and help, thank you!
In recent years, important progress has been made in the study of landscape beauty and perceptual experience in the fields of urban planning[1-3], environmental design[4-6], tourism planning[7,8], and psychology[9-12].
- Qin, X.C.; Fang, M.J.; Yang, D.X.; Wangari, V.W. Quantitative evaluation of attraction intensity of highway landscape visual elements based on dynamic perception. Environmental Impact Assessment Review 2023, 100, doi:10.1016/j.eiar.2023.107081.
- Chen, G.D.; Sun, X.Y.; Yu, W.B.; Wang, H. Analysis Model of the Relationship between Public Spatial Forms in Traditional Villages and Scenic Beauty Preference Based on LiDAR Point Cloud Data. Land 2022, 11, doi:10.3390/land11081133.
- Li, J.; Zhang, Z.H.; Jing, F.; Gao, J.; Ma, J.Y.; Shao, G.F.; Noel, S. An evaluation of urban green space in Shanghai, China, using eye tracking. Urban Forestry & Urban Greening 2020, 56, doi:10.1016/j.ufug.2020.126903.
- Wang, Y.; Yang, G.F.; Lu, Y.J. Evaluation of urban wetland landscapes based on a comprehensive model - a comparative study of three urban wetlands in Hangzhou, China. Environmental Research Communications 2023, 5, doi:10.1088/2515-7620/acbf12.
- Lopez-Martinez, F. Visual landscape preferences in Mediterranean areas and their socio-demographic influences. Ecological Engineering 2017, 104, 205-215, doi:10.1016/j.ecoleng.2017.04.036.
- Luckmann, K.; Lagemann, V.; Menzel, S. Landscape Assessment and Evaluation of Young People: Comparing Nature-Orientated Habitat and Engineered Habitat Preferences. Environment and Behavior 2013, 45, 86-112, doi:10.1177/0013916511411478.
- Sun, M.H.; Zhang, X.Y.; Ryan, C. Perceiving tourist destination landscapes through Chinese eyes: The case of South Island, New Zealand. Tourism Management 2015, 46, 582-595, doi:10.1016/j.tourman.2014.08.010.
- Akten, M.; Celik, M. Evaluation of visual landscape perception for Incilipinar and Adalet Park cases. Journal of Food Agriculture & Environment 2013, 11, 1532-1538.
- Yazici, K.; Asur, F. ASSESSMENT OF LANDSCAPE TYPES AND AESTHETIC QUALITIES BY VISUAL PREFERENCES (TOKAT, TURKEY). Journal of Environmental Protection and Ecology 2021, 22, 340-349.
- Jacques, D. Neuroaesthetics and landscape appreciation. Landscape Research 2021, 46, 116-127, doi:10.1080/01426397.2020.1832204.
- Bao, Y.; Yang, T.X.; Lin, X.X.; Fang, Y.; Wang, Y.; Poppel, E.; Lei, Q. Aesthetic Preferences for Eastern and Western Traditional Visual Art: Identity Matters. Frontiers in Psychology 2016, 7, doi:10.3389/fpsyg.2016.01596.
- Tang, I.C.; Sullivan, W.C.; Chang, C.Y. Perceptual Evaluation of Natural Landscapes: The Role of the Individual Connection to Nature. Environment and Behavior 2015, 47, 595-617, doi:10.1177/0013916513520604.
In all, I found the reviewer’s comments are quite helpful, and I revised my paper point-by-point. Thank you and the review again for your help!
Sincerely yours,
Yue Chen, Qikang Zhong, Bo Li.
Reviewer 2 Report
The study firstly discusses the inaccuracy of single viewpoint photographs in the evaluation of landscape effect, and then analyzes the influencing factors of landscape effect by evaluating the panoramic photo collection and overall sense of beauty (OSBS) of 20 spatial scenes of the Yaozijian Ancient Road in Anhua County. The basic research of the article is sufficient, a full literature review has been done, and the experimental method is reasonable. However, the article has the following problems:
1.The background and significance of the study need to be explained in more detail so that readers can better understand the motivation and contribution of the study.
2、The main findings and methods of previous studies on visual perception of landscape need to be elaborated in more detail so that readers can understand the position of the study in the field.
3. the discussion section needs to explore in more depth the implications and possible interpretations of the research findings. Some hypotheses can be proposed and discussed as to why positive and negative perspectives have different effects on overall landscape aesthetics.
4, Whether there is any data or research to support the fact that the main tourist and consumer in the village are young people or professional students. It will affect the issue of the selection of the group of subjects in the evaluation test.
5, the end of the article discusses the degree of influence of each spatial factor on the OSBS and the reasons, the mechanism of its influence can be combined with the index of the attributes of the elements of the quantification of the relevant analysis, the current text of the mechanism of the influence of the expression of the slightly insufficient basis.
Some expressions in the article are not clear, and it is suggested to proofread in native language.
Author Response
Dear reviewer:
Thanks very much for taking your time to review this manuscript. We appreciate all your generous comments and suggestions! Please find my itemized responses in below and my revisions in the re-submitted files.
- Response to comment:The background and significance of the study need to be explained in more detail so that readers can better understand the motivation and contribution of the study.
Response: Thank you for your detailed comments. We attach great importance to this comment. We have reinserted the research background and significance into the first sentence of the abstract, and in the introduction, we have provided a more detailed explanation of the research background and significance. We hope that our modifications meet with your approval. Thank you for your guidance.
- Response to comment:The main findings and methods of previous studies on visual perception of landscape need to be elaborated in more detail so that readers can understand the position of the study in the field.
Response: Thank you for your detailed comments. We attach great importance to this comment. We have deepened our reading of the literature in the field of landscape visual perception, comprehended and summarized it, and incorporated it into the introduction so that readers can understand the position of the study in the field.
- Response to comment: the discussion section needs to explore in more depth the implications and possible interpretations of the research findings. Some hypotheses can be proposed and discussed as to why positive and negative perspectives have different effects on overall landscape aesthetics.
Response: Thank you for your detailed comments. We attach great importance to this comment. We have delved deeper into the implications of the research findings and explored possible explanations. Furthermore, we discussed why positive and negative viewpoints have different effects on overall landscape aesthetics. Once again, thank you for your professional guidance, and we greatly value the opportunity to benefit from your expertise.
- Response to comment:Whether there is any data or research to support the fact that the main tourist and consumer in the village are young people or professional students. It will affect the issue of the selection of the group of subjects in the evaluation test.
Response: Thank you for your detailed comments. We have added relevant literature and citations, hoping to contribute to this study.
- Response to comment:the end of the article discusses the degree of influence of each spatial factor on the OSBS and the reasons, the mechanism of its influence can be combined with the index of the attributes of the elements of the quantification of the relevant analysis, the current text of the mechanism of the influence of the expression of the slightly insufficient basis.
Response: Thank you for your detailed comments. We attach great importance to this comment. Perhaps due to unclear language expression and the fact that English is not our native language, it has led to difficulties in reading the article and unclear statements. In response to this, we have conducted an overall language refinement, hoping for your understanding and seeking your assistance for the publication of this paper. Thank you very much.
- Response to comment:Some expressions in the article are not clear, and it is suggested to proofread in native language.
Response: We apologize for the unclear expression in the article due to our non-proficiency in the English language as native speakers. To address this, we have sought the assistance of more proficient native speakers for language revisions, aiming to make the presentation of this paper clearer. We sincerely apologize for any inconvenience caused.
In all, I found the reviewer’s comments are quite helpful, and I revised my paper point-by-point. Thank you and the review again for your help!
Sincerely yours,
Yue Chen, Qikang Zhong, Bo Li.
Reviewer 3 Report
Dear Authors,
Congratulations on your research. I have found your manuscript consistent, precise, and scientifically sound. Therefore, I only have a few minor suggestions.
Firstly, when you describe the advantages of panoramas as compared to 2D pictures, it would be essential to note how they were experienced by the respondents - as a movie seen on a screen, with the use of VR headsets, or any other means.
Subsequently, I wasn't sure whether my understanding of positive and negative viewpoints was the same as yours. I suggest describing those terms, maybe providing examples.
I would also like to raise a question regarding criterion C13 - The type of vegetation. There can be different types of vegetation - some types might have a positive impact, and some other types can have a negative impact on scenic beauty perception. I suggest defining this criterion better, at least at the moment it is referred to in the discussion part. "Type of vegetation" might be differentiated - then how do you treat it as one criterion across your research?
Last but not least, I would suggest developing conclusions on how your findings have important implications for landscape design, urban planning, and environmental management.
The use of the English language is good, I didn't detect any significant problems. However, your paper might still benefit from proofreading, as there were some sentences that lacked interpunction or seemed to blend with the following one.
Best regards,
Author Response
Dear reviewer:
Thanks very much for taking your time to review this manuscript. We appreciate all your generous comments and suggestions! Please find my itemized responses in below and my revisions in the re-submitted files.
- Response to comment:Firstly, when you describe the advantages of panoramas as compared to 2D pictures, it would be essential to note how they were experienced by the respondents - as a movie seen on a screen, with the use of VR headsets, or any other means.
Response: Thank you for your detailed comments. We attach great importance to this comment. We have appropriately added this part of the content in section 2.1, hoping that the revised content will meet with your approval. Once again, thank you for your professional guidance.
- Response to comment:Subsequently, I wasn't sure whether my understanding of positive and negative viewpoints was the same as yours. I suggest describing those terms, maybe providing examples.
Response: Thank you for your detailed comments. We attach great importance to this comment. We have already explained and clarified this in the introduction. Additionally, in section 3.4.3, we have also included photos illustrating positive and negative viewpoints.
- Response to comment:I would also like to raise a question regarding criterion C13 - The type of vegetation. There can be different types of vegetation - some types might have a positive impact, and some other types can have a negative impact on scenic beauty perception. I suggest defining this criterion better, at least at the moment it is referred to in the discussion part. "Type of vegetation" might be differentiated - then how do you treat it as one criterion across your research?
Response: Thank you for your detailed comments. We attach great importance to this comment. Based on your feedback, we have redefined the categories according to the quantity, aiming for a better assessment of this criterion. We hope that this revised standard will be more understandable for readers. Once again, thank you for your professional guidance; you are truly excellent. Regarding the impacts of different types of vegetation, it is an aspect that our study can consider in the future. Currently, we have only taken into account the diversity of the vegetation.
- Response to comment:Last but not least, I would suggest developing conclusions on how your findings have important implications for landscape design, urban planning, and environmental management.
Response: Thank you for your detailed comments. We attach great importance to this comment. We briefly discussed this part and placed it in the conclusion section.
- Response to comment:The use of the English language is good, I didn't detect any significant problems. However, your paper might still benefit from proofreading, as there were some sentences that lacked interpunction or seemed to blend with the following one.
Response: Thank you for your detailed comments. We attach great importance to this comment. Due to unclear language expression and the fact that English is not our native language, it has led to difficulties in reading the article. In response to this, we have conducted an overall language refinement, hoping for your understanding and seeking your assistance for the publication of this paper. Thank you very much.
In all, I found the reviewer’s comments are quite helpful, and I revised my paper point-by-point. Thank you and the review again for your help!
Sincerely yours,
Yue Chen, Qikang Zhong, Bo Li.
Reviewer 4 Report
The overall assessment of the article is positive, as all of the content already included is unobjectionable in terms of content and language. On the other hand, it would be worthwhile to supplement the paper with some further issues on three points presented below:
1. Firstly, in the Review chapter, the authors refer to previous research, but only concerning research techniques. In contrast, a brief review of research in the field of landscape perception itself and the links between different landscape elements and landscape evaluation is missing.
2. Secondly, there is a lack of reference of the results obtained by the authors precisely to other studies conducted on similar topics.
3. Finally, in the final section of the paper, in addition to a general statement about the practical usefulness of the results obtained, it would be useful to write more concretely for which purposes, which institutions and to what extent these results can be used.
Author Response
Dear reviewer:
Thanks very much for taking your time to review this manuscript. We appreciate all your generous comments and suggestions! Please find my itemized responses in below and my revisions in the re-submitted files.
- Response to comment:Firstly, in the Review chapter, the authors refer to previous research, but only concerning research techniques. In contrast, a brief review of research in the field of landscape perception itself and the links between different landscape elements and landscape evaluation is missing.
Response: Thank you for your detailed comments. We attach great importance to this comment. We highly value your professional opinion, and we have also included the missing content. However, we believe that it is more appropriate to place it in the introduction section. Therefore, please refer to the introduction chapter for the specific details.
- Response to comment:Secondly, there is a lack of reference of the results obtained by the authors precisely to other studies conducted on similar topics.
Response: Thank you for your detailed comments. We attach great importance to this comment. The theme of this article is the innovative aspect of this research. As such, there have been relatively few studies with similar themes, and the available literature for reference is limited. Therefore, we have corroborated our findings through other relevant research. We apologize for our limited capabilities. Moving forward, we will intensify our efforts to study and research in this thematic area, hoping to continue benefiting from your professional guidance in the future.
- Response to comment:Finally, in the final section of the paper, in addition to a general statement about the practical usefulness of the results obtained, it would be useful to write more concretely for which purposes, which institutions and to what extent these results can be used.
Response: Thank you for your detailed comments. We attach great importance to this comment. Due to the word limit in the conclusion section, we have briefly added the application of these findings in urban planning, landscape design, and related professions. Once again, thank you for your professional guidance, and we hope you continue to have a successful academic career and remain happy every day!
In all, I found the reviewer’s comments are quite helpful, and I revised my paper point-by-point. Thank you and the review again for your help!
Sincerely yours,
Yue Chen, Qikang Zhong, Bo Li.